# New Markers of Renal Failure in Multiple Myeloma and Monoclonal Gammopathies

**DOI:** 10.3390/jcm9061652

**Published:** 2020-05-31

**Authors:** Karolina Woziwodzka, David H. Vesole, Jolanta Małyszko, Krzysztof Batko, Artur Jurczyszyn, Ewa Koc-Żórawska, Marcin Krzanowski, Jacek Małyszko, Marcin Żórawski, Anna Waszczuk-Gajda, Marek Kuźniewski, Katarzyna Krzanowska

**Affiliations:** 1Departament of Nephrology, Jagiellonian University Medical College, 31-008 Cracow, Poland; woziwodzka.karolina@gmail.com (K.W.); batko.krzysztof@gmail.com (K.B.); mkrzanowski@op.pl (M.K.); marek.kuzniewski@uj.edu.pl (M.K.); 2John Theurer Cancer Center, Hackensack University Medical Center, Hackensack, NJ 07601, USA; david.vesole@hackensackmeridian.org; 3Department of Nephrology, Dialysis and Internal Medicine, Warsaw Medical University, 02-091 Warszawa, Poland; jolmal@poczta.onet.pl; 4Departament of Hematology, Jagiellonian University Medical College, 31-008 Cracow, Poland; mmjurczy@cyf-kr.edu.pl; 5Second Department of Nephrology and Hypertension with Dialysis Unit, Medical University of Bialystok, 15-276 Białystok, Poland; ewakoczorawska@wp.pl; 6Departament of Nephrology, Medical University, 15-089 Bialystok, Poland; jacek.malyszko@umb.edu.pl; 7Department of Clinical Medicine, Medical University of Bialystok, 15-254 Bialystok, Poland; mzorawski@wp.pl; 8Department of Hematology, Oncology and Internal Diseases, Medical University of Warsaw, 02-091 Warsaw, Poland; annawaszczukgajda@gmail.com

**Keywords:** biomarkers, kidney injure, monoclonal gammopathies, multiple myeloma

## Abstract

Multiple myeloma (MM) is a common plasma cell malignancy, which is responsible for significant mortality, often related to severe renal impairment (RI). Kidney injury can limit therapeutic choices and may often translate into poor outcomes, but it remains potentially reversible in a proportion of patients. The most accessible, conventional markers of RI are subject to several shortfalls, among which are the delayed onset following kidney insult, multiple interfering factors, and lesser sensitivity to mild changes in glomerular filtration. Neutrophil gelatinase-associated lipocalin (NGAL) and cystatin C have accumulated large interest in MM-RI due to being very sensitive markers of renal injury, as well as indicators of tubular-glomerular axis impairment. Of interest, recent data suggest that prediction of acute kidney injury may be aided by urinary tissue inhibitor of matrix metalloproteinase-2 (TIMP-2) and insulin-like growth factor-binding protein 7 (IGFBP7), which both act to induce G1 cell cycle arrest, reflective of a state of pre-injury, and thus may be superior to other measures of kidney insult (NGAL, kidney injury molecule ((KIM-1)). Moreover, TIMP-2 seems to be a biomarker dedicated to distal tubular cells, whereas insulin-like growth factor-binding protein 7 (IGFBP7) secretion has been found in proximal tubule cells. IGFBP7 can also identify a subsection of the normal proximal nephron, even, maybe the one that is responding to insult. They may be adopted into a conceptual screening panel for MM-RI. Unfortunately, no biomarker is ideal (influence of non-renal, biologic factors), and novel measures are limited by economic constraints, availability, lack of standardization. With the emergence of more advanced diagnostic and prognostic MM models, markers reflective of disease processes (including RI) are of high interest. Candidate molecules also include peptidome markers.

## 1. Multiple Myeloma and Renal Impairment—An Overview

In the United States, it has been estimated that multiple myeloma (MM), a plasma cell malignancy, will account for nearly 13,000 deaths in 2019, while over 32,000 new cases will be diagnosed [1]. Monoclonal gammopathy of undetermined significance (MGUS) is a common, asymptomatic condition, which may often precede MM, and is found in about 3% of individuals at or over the age of 50 [2]. MM is characterized by a plasma cell clone, which releases non-functional monoclonal proteins (e.g., immunoglobulins, or parts thereof), which can be found in the serum and/or urine in the majority of cases. The most commonly detected immunoglobulin (Ig) is IgG with its subtypes (52%) and IgA (21%), whereas light chain secretion is found in 16% of patients [3]. 

There is an ongoing search for more reliable biomarkers of end-organ involvement in MM [4]. Renal impairment (RI) is one of the cardinal features of MM. Nearly half of newly-diagnosed MM patients may have some degree of RI at diagnosis, though it should be kept in mind that large variability is prevalent across studies, which likely depends on the chosen RI measure and criterion [3]. In a study of newly diagnosed MM patients, 31% of 1353 cases were observed with elevated serum creatinine, while severe renal impairment was present in 16% [5]. When estimating creatinine clearance, 49% of patients were determined to have a degree of renal failure at diagnosis. Definitions of RI are subject to several shortfalls, e.g., individual cohort characteristics and use of equations extrapolated from chronic kidney disease (CKD), which may not always yield equivalent results in MM [6]. 

Monoclonal Ig-related kidney disease occurs in three main forms; the most common cast nephropathy (i.e., myeloma kidney), monoclonal Ig deposition disease (MIDD), and amyloid light-chain (AL) amyloidosis. Table 1 provides an outlook on the most common renal disorders associated with MM, with regard to the underlying mechanism and clinical presentation (based on [7,8,9,10]). In an analysis of 77 autopsies of patients dying from plasma cell malignancy complications, heterogeneity of kidney pathology was observed, with cast nephropathy as the main lesion [11]. IgM clone-related kidney complications are considered to be rare, though interestingly, a variety of kidney features is observed without association with the underlying type of hematologic disorder [12]. Renal manifestations secondary to monoclonal gammopathy or immune cell dysfunction, which do not fulfill MM criteria, should be promptly diagnosed with kidney biopsy (monoclonal gammopathy of renal significance (MGRS)) and treated to control the offending clone [13]. Comprehensive reviews on MM Ig-related kidney pathology are available elsewhere [7,10].

The pathogenesis of renal injury in MM is multi-factorial, which reflects the heterogeneity of the primary disease, with variable composition and properties of monoclonal paraproteins (a simplified overview, see Table 2) [8]. Several kidney features may predispose this particular organ to be affected by circulating paraproteins—an abundance of renal capillaries with large blood flow, the concentration of filtered solutes, and the presence of a variety of exogenous and endogenous substances [14]. Tumor burden is another important factor. Excess free light chain (FLC) filtration and overburdening of proximal resorptive capacity, with subsequent uromodulin interactions in distal tubules, may precipitate cast formation, inducing tubulointerstitial damage and fibrogenesis [8]. Monoclonal proteins can also deposit along tubular or basement membranes, leading to particular histologic patterns [8]. The proportion of patients with RI is variable in the MM spectrum when categorized according to the M component [5]. Interestingly, not all patients with high levels of FLCs present with RI [15]. Physicochemical properties of the secreted paraprotein may determine pathological features, for which a variety of Ig-dependent and independent mechanisms have been described (see Figure 1) [8,14]. However, it has to be emphasized that FLC levels >800 mg/L are good predictors of severe RI, regardless of the paraprotein type [16]. Tools to recognize the degree of nephrotoxicity for particular monoclonal paraproteins are still lacking; therefore, very early identification of renal insult is crucial. It has been shown that renal failure is reversible for a substantial proportion of MM patients, while reversal may benefit long-term survival [17].

Severe renal failure, in particular, dialysis-dependency, remains a deleterious organ complication with an increased risk of early death and a poor prognosis [18,19]. A timely diagnosis is necessary to prevent poor outcomes. It should be noted that effectively reducing tumor burden and preventing the production of the offending paraprotein is the underlying principle for a durable response. Improving the clearance of FLCs will only be effective if clonal production is reduced, though a direct relationship between FLCs and RI severity has recently been underscored [16]. Retrospective studies have previously suggested that in addition to novel chemotherapeutic agents, high-cut-off hemodialysis (HCO) may lead to renal recovery and improved survival in dialysis-dependent acute renal failure [20]. Subsequently, the importance of early FLC reduction was shown to be associated with renal recovery (60% reduction of FLCs at 3 weeks led to a renal recovery in 80% of cases) and supported by larger series [21,22]. Case-control studies have also indicated the benefit of HCO over conventional hemodialysis (HD) [23]. However, randomized controlled trials (namely, NCT01208818 (MYRE) and NCT007005321 (EuLITE) have raised some controversies with a significant reduction in FLC levels, but no significant difference in the primary endpoints (dialysis independence at 3 months), though further analyses suggested improvements at later time points, while the study itself may have been underpowered [24,25].

Reversal of renal injury is particularly significant since the presence of kidney disease limits therapeutic options and makes stem cell transplant more challenging [26]. The number of patients with MM-related end-stage chronic kidney disease on renal replacement therapy (RRT) has increased during the last twenty years, particularly with more treatments becoming available and patients living longer [27]. Indeed, data suggest that novel therapies (proteasome inhibitors, immunomodulatory agents, monoclonal antibodies) improve RI and survival [28]. Bortezomib-based regimens, particularly triplet combinations, maybe favorable in achieving a renal response, dialysis independence, and improving survival [29,30]. Recently, the French national registry on RRT in end-stage renal disease (without RRT for AKI (acute kidney injury)) was analyzed between 2002 and 2011 to determine the trend and survival of patients with monoclonal gammopathies on chronic dialysis. Although prognosis still remains poor, improvement in survival was noticed after 2006, which might be attributed to renal recovery following the changes in the treatment armamentarium (e.g., bortezomib) [31]. It was noted that kidney transplantation still remains a rare therapeutic measure in myeloma cast nephropathy, as well as light-chain deposition disease, particularly with the risks of recurrence, Ig-mediated graft dysfunction, and infection counterbalancing the potential benefits. However, with the advent of risk-adapted therapy, prognostic models integrating cytogenetic and clinical features of disease may identify patients with an “optimal” phenotype for transplantation. It has been proposed that the selection of “low risk” patients (absence of adverse indices of disease) might change the outlook on a kidney transplant in MM and shift the dismal implications of end-stage renal disease [32].

## 2. Predictive and Prognostic Tools in Myeloma-Related Kidney Disease

Personalized medicine is a new and emerging field that will hopefully become a standard of care. The concept relies on our understanding of pathophysiology, alongside the ability to measure various aspects of organ/system function in order to tailor management to the individual. Risk stratification algorithms are currently being developed for clinical scenarios (not only at diagnosis but also the initiation of second-line treatment) and involve indices of the individual and disease, e.g., aggressive character and frailty [33]. Conceptual models for the relationships between disease driving processes, patient characteristics, and key outcomes have been constructed by expert panels and aided by real-world evidence, providing a framework for future research [34]. Recently, a foundational model has been constructed for processes involved in myeloma-related renal impairment (focusing on proximal tubule injury by FLCs inducing an inflammatory response) [35]. Relevant biomarkers for kidney injury and myeloma burden are needed for the conceptualization and development of a more complete model [35]. There is a lot of promise in kidney function models, which would account for disease-defining processes. Early identification of subclinical renal damage (by conventional measures), prediction of renal response, or identifying MGUS at risk of MGRS are future avenues of research that could shape best practice. In the case of novel kidney injury indicators, two overlapping entities have to be considered: (1) acute kidney injury, for which early identification is crucial, and (2) CKD, in which there is a necessity for frequent, reliable assays. 

Serum creatinine (SCr) remains a staple of practice, though its interpretation requires awareness of potential confounding factors, which remains a general principle to be kept in mind for all biomarkers. Several shortfalls have been outlined in the literature: (1) elevation in prerenal azotemia with no tubular insult, (2) production rate dependent on individual and clinical characteristics, i.e., age, gender, muscle mass, and medication use, (3) late rise in up to 72 h following injury, (4) large “renal reserve” [36,37]. In several conditions (e.g., sepsis, liver disease, muscle wasting), SCr may not reflect the actual fall in glomerular filtration [38]. Knudsen et al. evaluated data from over 1200 MM patients and showed that renal impairment was identified by creatinine clearance in 49% of patients, as compared to abnormal serum values in 31%, which indicated a more favorable discriminatory value of the urine assay [5]. Within the last few years, several candidate markers have emerged in nephrology, which appear to be valuable for study and validation in MM-related kidney disease. The actual fall in glomerular filtration may not be evident initially in routine tests, while more reliable assessments are significant for dose adjustments and decisions to initiate treatment, which is where markers sensitive for early injury may prove particularly useful.

## 3. Markers of Myeloma Burden of Renal Significance

From an empirical standpoint, constructing a model reflective of MM-related kidney disease should account for tumor burden and the dynamic of processes involved in direct kidney damage. FLCs are a valuable and well-established tool. It should be kept in mind that in renal failure, the FLC ratio may be elevated above the usual norm (0.26–1.65), and the diagnostic range should be extended [39,40,41]. Among the 1027 newly diagnosed MM patients reviewed by Kyle et al., increased SCr of more than 2 mg/dL was found only in 19% of examined cases, whereas abnormal beta2-microglobulin level (β2M) was detected in 75%, alongside light chains in urine among 78% [3]. Elevated markers reflecting tumor burden, such as β2M, should be considered in the context of both clinical phenotype and kidney involvement. β2M is an 11.8 kDa molecule, which has been studied as part of the neonatal Fc receptor (FcRn)–β2M axis exerting regulatory effects on albumin, iron availability, and IgG, which may play a role in immune-mediated renal disease. It has been observed that β2M may reflect glomerular and tubular disorders, with performance similar to Cr-based formulas for renal function. Urinary and plasma assay has also been outlined as advantageous in being reliable and cost-effective [42]. However, coexisting morbidity with high cell turnover may limit the use of this marker in screening, particularly in an elderly population. 

These inherent problems with existing biomarkers are the rationale behind novel investigations of new markers of RI. An ideal biomarker should be readily assessable, inexpensive, universally available and would aid in differentiating the cause of renal failure, show the etiology and site of damage (e.g., proximal or distal tubule, glomeruli), predict the severity of the injury, need for RRT, and allow monitoring of the effects of treatment. Infrequent use of the currently available biomarkers is the result of expensive and complicated methods of their measurement and lack of universally accepted and validated standards. In the next few paragraphs, we have described the most promising biomarkers in RI related to MM and their utility. Characteristics of prominent, well-investigated biomarkers are shown in Table 3. 

## 4. Tubular-Glomerular Axis Impairment

### 4.1. Neutrophil Gelatinase-Associated Lipocalin—A Promise for Real-Time Monitoring of Tubular Injury?

Neutrophil gelatinase-associated lipocalin (NGAL) is a small glycoprotein, existing in several molecular forms and expressed in various tissues, for which the assay in urine is of particular interest in suspicion of acute injury, as it is filtered and proximally reabsorbed, while its production occurs local to injured tubular cells [48,49]. However, it may also be released by activated neutrophils, potentially limiting utility in several instances of critical care (inflammation and sepsis), with comprehensive reports of heterogeneous performance in predicting RRT [48]. There are several characteristic properties of NGAL: (1) detection within a few hours, (2) short half-life, (3) fast renal elimination from circulation, and (4) sensitivity in acute and chronic settings [50]. Lately, an appraisal of evidence on NGAL dysregulation in cancer has described the concept of NGAL overexpression resulting from hypoxic and inflammatory stimuli (potentially via active nuclear factor kappa-light-chain-enhancer of activated B cells (NF-κB) and mitogen-activated protein kinase (MAPK) pathways) originating from the tumor microenvironment. Interestingly, the activation of these pathways plays a significant role in both myeloma pathogenesis and related kidney lesions (induced by excess FLC processing) [8,51]. Indeed, in MM patients, serum/plasma NGAL levels share a relationship with M protein levels, international staging system (ISS) stage, and disease status, indicating utility as a bimodal marker of tumor burden and renal injury [50,52,53]. NGAL has been examined as a clinical marker in several clonal disorders (particularly in chronic myeloid leukemia (CML), with a purported link with tyrosine kinase breakpoint cluster region-Abelson (BCR-ABL)), though its predictive and prognostic value is still uncertain [54]. In 46 patients with MGUS compared with age and sex-matched controls, urine but not serum NGAL was considered a sensitive kidney injury marker [49]. In contrast, serum NGAL was shown to be elevated in all stages of the myeloma spectrum (MGUS, smoldering MM (SMM), and MM), whereas serum cystatin C was increased in symptomatic MM patients [50]. Malyszko et al. previously described the relationship of NGAL in serum, urine, and ultrafiltrate with the modality of RRT, renal function, and indices of inflammation [55].

Papassotiriou et al. observed that serum NGAL levels correlated significantly with cystatin C, which indicated an impairment of the tubular-glomerular axis, which was further supported by the associations with estimated glomerular filtration rate (eGFR) according to modification of diet in renal disease (MDRD) formula, and even more strongly with cystatin C-based equations. Following the forest fire theory (NGAL production by inflamed tubular cells rather than impaired clearance), it has been suggested that a combination of NGAL and cystatin C would reflect both active renal insult (serum NGAL) and functional nephron loss (serum cystatin C). Potential confounding by myeloma-derived inflammation has not been supported by correlations with indices of inflammation (interleukin-6 (Il-6) and C-reactive protein (CRP)), though circulating markers may not reflect local tissue levels [50,56].

Du et al. [53] examined a panel of renal biomarkers in MM patients with CKD, MM with normal kidney function, and healthy volunteers, observing elevated NGAL levels in both serum and urine in the “MM-renal group”, as compared to the remaining two. The positive predictive value of serum NGAL for MM with renal damage was marginally higher (86% vs. 72%) than the urine form, which might be preferable in discriminating this unique group of patients. However, as an indicator of kidney injury, urinary NGAL was more sensitive than its serum assay and was significantly associated with eGFR.

Chae at al. examined 199 samples from an equal number of MM patients (majority during follow-up visits) representing a heterogeneous group (*~*40% complete response, 30% progressive disease). Several predictors (eGFR, M protein, and cystatin C) were identified for NGAL, which itself correlated with CKD according to kidney disease improving global outcomes (KDIGO) in MM patients and markers of myeloma burden [52]. Serum assay may be potentially more versatile if the urine assay requires 24-h urine collection for reliability.

### 4.2. Kidney Injury Molecule-1 (KIM-1) 

Kidney injury molecule-1 (KIM-1) is a urinary molecule utilized for detecting drug-related proximal tubule damage. Its potential in the early identification of developing AKI, as well as the risk of progression once the injury has already occurred, has justified large clinical interest. Dimopoulos et al. reported the only data currently available on its application in the setting of myeloma, concluding very low utility due to differential pathogenesis of renal dysfunction (as opposed to drug-induced nephrotoxicity) [57,58].

### 4.3. Cystatin C

Cystatin C (Cys-C) is a positively charged, low molecular weight protein (13 kDa), which acts as a cysteine protease inhibitor, and is released from all nucleated cells, with its production rate relatively stable (this changes at 50–60 years of age). Due to its free glomerular filtration and near-complete tubular catabolism (combined with a lack of secretion), urinary presence may indicate a tubular injury. Physiologically, specimens for its measurement include serum and plasma, with its concentration proportional to that of glomerular filtration due to constant release. It has also been attributed with several important advantages: (1) sensitivity to small GFR change, whereas creatinine might still remain within the reference range, (2) lack of circadian alterations, which allow for single urine specimen assay, (3) short half-life and extracellular distribution, which rapidly elevate serum concentrations, (4) stability in some inflammatory and metabolic conditions, (5) less variability due to several individual characteristics (age, sex, race, muscle mass) [37,59,60,61]. On the other hand, Cys-C production may be altered under certain conditions of increased metabolic rate (potentially through increased cell turnover), e.g., malignancy, thyroid dysfunction, and steroid use [62]. Recent meta-analyses have shown that cystatin C can be favorable alone or combined with creatinine to predict mortality risk and end-stage renal disease (ESRD) in various populations [63].

Early studies in MM have suggested that although Cys-C is more sensitive in identifying moderate GFR changes than SCr, the use of conventional eGFR equations may be preferable due to a lack of superiority and expensive assay. Subsequent studies have shown that Cys-C-based chronic kidney disease epidemiology collaboration (CKD-EPI) equations are able to discriminate more MM patients (≤70 years of age) with stage 3–5 RI than SCr-based formulas (both MDRD and SCr-CKD-EPI). Moreover, the use of Cys-C CKD-EPI has been shown to reclassify nearly 30% of patients to a higher CKD stage (from MDRD). Finally, in the multivariate analysis of survival, Cys-C (but not SCr)-based equations for eGFR have been shown to be an independent predictor of survival, potentially due to the relationship with myeloma biology [62]. There is now an array of evidence that suggests Cys-C is a very sensitive marker of RI in MM [50,62,64]. Terpos et al. previously studied 157 newly diagnosed MM patients (pre and post bortezomib) compared with 52 healthy controls. Cys-C showed a significant relationship with the ISS stage, advanced bone disease, and independent prediction of survival, which was also supported in the construction of a prognostic model [65]. In another cohort of 68 patients after high-dose melphalan and autologous stem cell transplantation, a gradual rise in Cys-C accordingly to tumor stage, as well as a relationship with poor prognosis, was observed [64]. At diagnosis, 35% of patients presented with elevated Cys-C, as compared to 16% for SCr. Close associations between Cys-C, SCr, and β2M have also been reported [65,66,67]. Gene expression profiling in microarray studies of MM has demonstrated that Cys-C is among the most highly upregulated genes [68]. This may indicate Cys-C shares a unique relationship with tumor burden and renal function.

### 4.4. Tissue Inhibitor of Matrix Metalloproteinase-2 (TIMP-2) and IGFBP-7

In patients enrolled in a randomized clinical trial, several cytokines and angiogenic factors (angiopoietin-2 (Ang-2), fibroblast growth factor-2 (FGF-2), hepatocyte growth factor (HGF), vascular endothelial growth factor (VEGF), platelet-derived growth factor-beta (PDGF-β), interleukin-8 (IL-8), tumor necrosis factor-alpha (TNF-α), tissue inhibitor of matrix metalloproteinase-1 (TIMP-1), and tissue inhibitor of matrix metalloproteinase-2 (TIMP-2)) were analyzed for a relationship with outcomes and response measures. Levels of all markers were markedly different from controls, while in the majority of patients, serum and bone marrow plasma levels were similar. This has often been considered a limitation of peripheral markers not reflecting the disease microenvironment and may be a promising finding, particularly due to the association of several of these molecules with therapeutic response [69]. Identification of disease subsets with a profile of peripheral cytokines and angiogenic molecules also poses the potential to differentiate patients who could benefit from additional target therapy or provide indices of refractory disease. Bone marrow mesenchymal stem cells may be a source of potential markers, with intrinsic overproduction of metalloproteinases (MMPs) and their inhibitors (TIMPs), potentially reflective of tumor burden and invasiveness [70]. Preliminary studies of bone remodeling markers identified TIMP-1 to be associated with bone lesions, advanced disease and poor survival [71].

Of interest, recent data suggest that prediction of acute kidney injury may be aided by urinary TIMP-2 and insulin-like growth factor-binding protein 7 (IGFBP7), which both act to induce G1 cell cycle arrest, reflective of a state of pre-injury, and thus may be superior to other measures of kidney insult (NGAL, KIM-1) [72]. Moreover, TIMP-2 seems to be a biomarker dedicated to distal tubular cells, whereas IGFBP-7 secretion has been found in proximal tubule cells. IGFBP7 can also identify a subsection of the normal proximal nephron, even, maybe the one that is responding to insult [73,74]. Yet, more studies should prove the role of TIMP-2 and IGFBP-7 as tubular injury molecules with differentiation to particular parts of tubes and because of their participation in the development of MM as biomarkers of tubules in MM.

### 4.5. N-Acetyl-β-Glucosaminidase (NAG) and Retinol-Binding Protein (RBP)

Among other biomarkers, retinol-binding protein (RBP) and lower N-acetyl-β-D-amino-glucosaminidase (NAG) have been considered useful in a subset of patients with MM and renal impairment presenting without conventional abnormalities, though these markers were not correlated with disease severity [74]. In a study of 278 patients with MM, low molecular weight urinary proteins were examined with higher urinary RBP and lower NAG in the glomerular damage group (as compared to tubular damage). A relationship with tubulointerstitial lesions was significant for urinary RBP, which was observed as a more favorable measure of renal insult, with higher specificity over NAG [75]. RBP has previously been identified as an independent risk factor for renal survival in biopsy-proven light chain deposition disease [76].

## 5. Novel Research Fields in the Search for Myeloma Markers

### 5.1. Prediction-Models and Artificial Intelligence

A shift in the research approach from a singular index biomarker to a panel best reflective of the complex interactions (linked to key outcomes) may aid in the search for novel markers. Utilizing gene arrays to identify a “molecular fingerprint” and proteomics to focus on specific, high-interest proteins is promising [77]. Studies have also examined metabolomic biomarkers, with the use of nuclear magnetic resonance spectroscopy and mass spectrometry tools [78,79,80]. The abnormal microenvironment in bone marrow is an ongoing focus of research into MM pathophysiology, implicated in both tumor initiation, progression, and refractory disease. Interactions between malignant plasma cells and the stroma include a variety of cytokines, chemokines, and growth and survival factors that may shape and explain, at least in part, the heterogeneity of the disease. Differentiating high-risk MGUS (for SMM/MM transformation), as well as a MM with a certain preference for organ involvement, may allow for initiating preventive measures and greater scrutiny in a specific group of patients.

As we are entering the era of “big data”, risk prediction models based on tools of artificial intelligence (machine learning) accrue more data as a sensitive tool, e.g., AKI prediction, which may provide valuable tools in the investigation of onco-nephrological entities [81]. Wang et al. investigated serum samples from a variable population of patients with different plasma cell dyscrasias, developing a diagnostic model via mass spectrometry technology, which showed some success in discriminating MM [82]. A more recent study by Bai et al. examined a heterogenous group of MM patients and demonstrated the use of supervised neural network analyses to differentiate MM from reference subjects, further identifying four candidate peptide biomarkers, which also showed a relationship with disease states (e.g., decrease after cytotoxic treatment) [83]. Similarly, Deulofeu et al. showed high sensitivity and specificity for the prediction of MM samples from healthy donors, utilizing mass spectrometry tools and artificial neural networks [80]. The potential of non-invasive measures that could rival invasive biopsy is certainly promising, but these findings still represent preliminary research. Yang et al. provided some preliminary results for the potential of serum peptidome analyses, with a diagnostic model of small peptides (less than 4000 Da) able to differentiate RI in MM [84].

### 5.2. Activin A

Activin A, a member of the transforming growth factor-beta (TGFB) superfamily, is involved in the regulation of cell inflammation, immunity, and cytokine cascades, while systemic activation of activin A signaling has also been linked to mineral bone disorders associated with chronic renal failure [85,86]. In patients with vasculitis, urinary activin A was able to distinguish patients with renal complications, which might be due to a relationship with the grade of renal disease [85]. It has also been shown that urinary activin A is elevated in patients from a spectrum of kidney disease, while it does not rise in healthy controls. In rodent models of ischemia-reperfusion, activin A expression is increased, while its mRNA is expressed in tubular cells of ischemic kidneys, and its concentration rises in urine 3 h after reperfusion. Urinary activin A is not present or present at very low levels in healthy controls and patients presenting with pre-renal causes of AKI [87]. Recently, data from patients with MM, MGRS, and MGUS have been reported, implicating the potential of activin A as a biomarker of RI [88]. Activin A was observed to be absent in healthy kidneys, while identified in a tubular cell in patients with MGRS, which led to a hypothesis of its tubular origin. The authors also observed a relationship with FLC concentrations, which might reflect a mechanism of paraprotein deposition with subsequent expression of tubular activin A. On the other hand, it has been proposed that as a 25-kDa molecule, its filtration and subsequent reabsorption might reflect its undetectable concentrations under physiological conditions, with a subsequent rise following tubular insult [88]. Further evaluation is necessary to clarify the potential of this marker.

## 6. Summary

Plasma cell dyscrasia can often manifest with renal damage due to the nature or amount of produced paraproteins (e.g., FLCs in cast nephropathy). Renal dysfunction may constrain the clinician in the choice of therapeutic measures (e.g., stem cell transplantation), which indicates the need for its early identification and limitation or prevention. Severe RI and massive proteinuria are associated with a lower likelihood of renal recovery. Due to the fact that RI is a negative predictor of overall survival and increases the risk of additional complications in MM patients, early detection is crucial for prompt implementation of adequate therapy that leads to disease control and reversal of renal injury. Renal biopsy is the most accurate method to determine histological lesions, but it cannot be used as a screening tool due to potential morbidity. This has led to the search for new biomarkers, which are non-invasive, sensitive, rapidly available, and related to the etiology of kidney disease. Moreover, early identification of patients with CKD may facilitate early interventions to reduce cardiovascular diseases or CKD progression. We are inclined to the view that a model or pattern comprised of serum and urine markers may be more precise than singular measures used so far. The use of peptidomics and novel analytical technology is a promising direction and should aid future efforts.

## Figures and Tables

**Figure 1 jcm-09-01652-f001:**
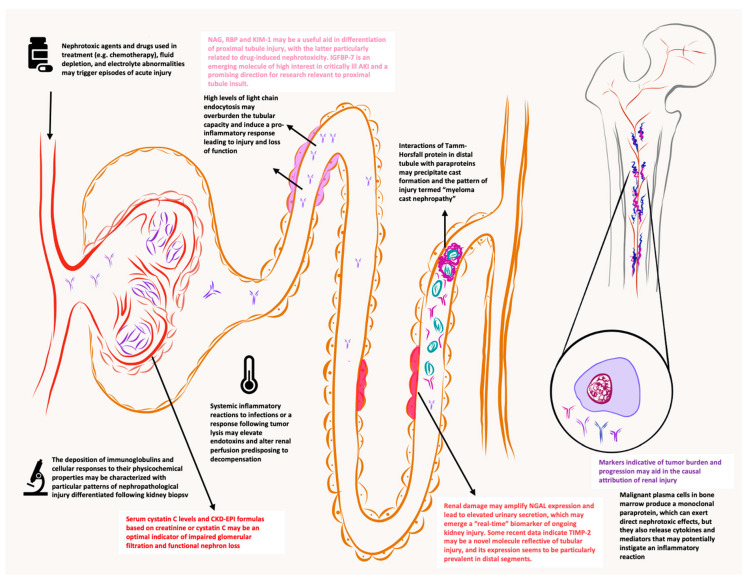
Proposed graphical representation of the nephron and malignant plasma cells in the bone marrow. Important mechanisms in the development of myeloma-related renal injury (monoclonal immunoglobulin dependent and independent) are discussed on sepia-colored panels, while novel renal biomarkers are identified in the proximity of the probable location of renal insult.

**Table 1 jcm-09-01652-t001:** An overview of the most common renal disorders in MM.

Renal Disease	Histopathologic Features	Clinical Features	Characteristics	Potential Exacerbating/Instigating Factors
Cast nephropathy	Eosinophilic, PAS (-) casts in tubules with the fractured appearance Reactive inflammatory infiltrate (giant cells)	AKI Proteinuria Low albuminuria CKD	The most common lesion in MM-related RI Light-chain only myeloma may be often associated with severe RI Other renal lesions may coexist	High tumor burden (>10 g/d light chain) Processing of excess paraprotein in proximal tubules may induce an inflammatory and fibrotic response Volume depletion sepsis Nephrotoxic agents (medications and contrast) Hypercalcemia Tumor lysis syndrome Rhabdomyolysis
Amyloidosis	Protein misfolding, fibril accumulation and deposits may occur in various kidney compartments (mainly glomerular) Monotypic IF staining Eosinophilic deposits with pale PAS and silver staining (+) Congo red stain, but normal tubular basement membrane thickness	Proteinuria Nephrotic Syndrome CKD	AL is the most common systemic amyloidosis (often λ-type) Other organs involvement may determine diagnostic (biopsy) and prognostic measures (cardiac) A proportion of patients with vascular-limited deposits (may present with lesser grade proteinuria) Often MGRS (low hematologic burden)	
MIDD	Nodular sclerosing Monotypic IF deposits in mesangium and along glomerular or tubular basement membranes (thickening) Powdery, electron-dense deposits	Proteinuria Hypertension CKD	Most often LCDD subtype (often *κ)* Other organs involvement possible (e.g., heart and liver) Often MGRS (low burden clonal production of paraprotein) Nodular sclerosing may be similar to diabetic nephropathy and AL amyloidosis patterns	

Abbreviations: acute kidney injury (AKI), immunoglobulin light chain amyloidosis (AL amyloidosis), chronic kidney disease (CKD), immunofluorescence (IF), light chain deposition disease (LCDD), periodic acid-Schiff (PAS), monoclonal gammopathy of renal significance (MGRS), multiple myeloma (MM), renal impairment (RI), monoclonal immunoglobulin deposition disease (MIDD).

**Table 2 jcm-09-01652-t002:** Overview of the potential mechanisms responsible for developing renal disorders in MM.

Potential Kidney Disease Triggers	General Effects	Clinical Features
Paraprotein with particular physicochemical properties (structure, solubility, charge, propensity for aggregation, protein interactions)	Potential nephrotoxicity directly or indirectly leading to glomerular, tubular, interstitial, and vascular lesions	Variable (immune dysregulation, host influences (e.g., tubular fluid characteristics, ion concentrations), or pathogenic interactions (e.g., disruption of the mesangial matrix) may result in different nephropathological presentations)
**Paraprotein characteristics**	**Main mechanism**	
Large molecular weight Ig or paraprotein	Limited ability to cross glomerular barriers leads to aggregation and deposition of Ig, which may induce local inflammation	Glomerulopathies
Low molecular weight Ig or paraprotein (e.g., light chains)	Filtration across glomerular barrier and interactions with proteins and tubular cells	Cast nephropathy or LCPT

Abbreviations: Ig, immunoglobulin; LCPT, Light chain proximal tubulopathy.

**Table 3 jcm-09-01652-t003:** Prominent or acclaimed biomarkers of renal impairment.

Biomarker	Sample	Comment	Time of Occurrence	Reference
Neutrophil gelatinase-associated lipocalin (NGAL)	urine, plasma	1. Initially identified bound to gelatinase in specific granules of the neutrophils but also may be induced in epithelial cells in the setting of inflammation or malignancy 2. Expression upregulated in kidney proximal tubule cells and urine following ischemic renal injury 3. An early indicator of AKI	Peak performance of urinary NGAL 3–12 h after renal injury	Vaidya et al. & Au et al. [43,44]
Kidney injury molecule-1 (KIM-1)	urine	1. Type-1 cell membrane glycoprotein upregulated in dedifferentiated proximal tubule epithelial cells 2. Elevated urinary levels are highly sensitive and specific for AKI after ischemia or nephrotoxins	Peak up to 24 h after injury, usually 3–6 h	Bonventre & Moledina et al. [45,46]
Cystatin C	urine, plasma	1. Important extracellular inhibitor of cysteine proteases 2. Filtered by the glomerulus and reabsorbed by proximal tubule cells 3. Elevated urinary levels reflect proximal tubular dysfunction; high levels may predict poorer outcome	The 24-h point after AKI might be a preferable selection	Vaidya et al. [43] & Yong et al. [47]
N-acetyl-β-glucosaminidase (NAG)	urine	1. Proximal tubule lysosomal enzyme 2. More stable than other urinary enzymes 3. Endogenous urea may inhibit the activity	Peak 12 h–4 days after kidney injury	Vaidya et al. [43]
Retinol-binding protein (RBP)	urine	1. Synthesized by the liver, involved in vitamin A transport 2. Filtered by the glomerulus and reabsorbed by proximal tubule cells 3. An early marker of tubular dysfunction		Vayda et al. [43]

Abbreviations: AKI, acute kidney disease; KIM-1, kidney injury molecule-1; NAG, N-acetyl-β-glucosaminidase; NGAL, neutrophil gelatinase-associated lipocalin; RBP, retinol-binding protein.

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
