# Peer review of "New Markers of Renal Failure in Multiple Myeloma and Monoclonal Gammopathies"

_jcm, 2020, doi:10.3390/jcm9061652_

Round 1
Reviewer 1 Report
This review paper focused on new biomarkers of renal failure in patients with multiple myeloma and monoclonal gammopathy. The authors summarized the common types of renal failure, the pathogenesis of renal failure, and new biomarkers for renal insufficiency. They recommended the use of these biomarkers for evaluation of kidney damage, but also discussed the limitation of these biomarkers in clinical practice.
Minor comments:
- The title and explanation are need in Figure 1.
- It is recommended that the text description of biomarkers be arranged in the order shown in Table 3.
- The title of section 5 is needed (page 11).
- The number of summary should be 6 (page 12).
Author Response
We thank the Reviewer for their valuable time and input. We hope to have addressed the comments to the best of our ability –
Figure 1 was redesigned and should be a comprehensive overview on the relevant pathomechanism and biomarkers to make the message clearer.
Following the advice given, biomarkers were reordered in Table 3 according to the sequence in the text, since we did not include all of the markers in the Table itself.
Finally, section 5 was termed as “novel research fields in the search for myeloma markers”.
Reviewer 2 Report
I read the review papers with great interest. The authors comprehensively reviewed the biological markers of renal impairment in multiple myeloma in this paper. The manuscript is valuable for physicians and myeloma experts. Just one major concern should be addressed. I think that figure 1 and the explanation is not satisfactory. So, the explanation regarding Ig-dependent and independent mechanisms of pathological process of renal impairment should be more clearly described in Figure 1.
Minor comments;
(1)in line 36 in page 1, KIM-1 should be spelled out.
(2)in Table 1 in page 3, what is "IF"? Add IF to abbreviations.
(3)in line 243 in page 9, KDIGO should be spelled out.
(4)in line 303 in page 10, make space between invasiveness and survival.
(5)in line 354 in page 11, TGFB should be spelled out.
Author Response
We thank the Reviewer for their valuable time and input. We hope to have addressed the comments to the best of our ability – Figure 1 was redesigned and should be a comprehensive overview on the relevant pathomechanism and biomarkers.
Minor comments;
(1)in line 36 in page 1, KIM-1 should be spelled out.
Done.
(2)in Table 1 in page 3, what is "IF"? Add IF to abbreviations.
Done – the relevant abbreviations have also been added.
(3)in line 243 in page 9, KDIGO should be spelled out.
Done.
(4)in line 303 in page 10, make space between invasiveness and survival.
Thank you, done.
(5)in line 354 in page 11, TGFB should be spelled out.
Done.